# The oomycete *Lagenisma coscinodisci* hijacks host alkaloid synthesis during infection of a marine diatom

Marine Vallet [1,8,9]*, Tim U.H. Baumeister [1,8], Filip Kaftan[2], Veit Grabe [3], Anthony Buaya[4], Marco Thines[4,5], Aleš Svatoš [2] & Georg Pohnert [1,6,7]*

Flagellated oomycetes frequently infect unicellular algae, thus limiting their proliferation. Here we show that the marine oomycete *Lagenisma coscinodisci* rewires the metabolome of the bloom-forming diatom *Coscinodiscus granii*, thereby promoting infection success. The algal alkaloids β-carboline and 4-carboxy-2,3,4,9-tetrahydro-1H-β-carboline are induced during infection. Single-cell profiling with AP-MALDI-MS and confocal laser scanning microscopy reveals that algal carbolines accumulate in the reproductive form of the parasite. The compounds arrest the algal cell division, increase the infection rate and induce plasmolysis in the host. Our results indicate that the oomycete manipulates the host metabolome to support its own multiplication.

[1] Research Group Plankton Community Interaction, Max Planck Institute for Chemical Ecology, Jena, Germany. [2] Research Group Mass Spectrometry/Proteomics, Max Planck Institute for Chemical Ecology, Jena, Germany. [3] Research Group Olfactory Coding, Department of Evolutionary Neuroethology, Max Planck Institute for Chemical Ecology, Jena, Germany. [4] Senckenberg Biodiversity and Climate Research Centre, Frankfurt am Main, Germany. [5] Department of Biological Sciences, Institute for Ecology, Evolution and Diversity, Goethe University, Frankfurt am Main, Germany. [6] Bioorganic Analytics, Institute for Inorganic and Analytical Chemistry, Friedrich Schiller University, Jena, Germany. [7] Microverse Cluster, Friedrich Schiller University Jena, Neugasse 23, 07743 Jena, Germany. [8] These authors contributed equally: Marine Vallet, Tim U. H. Baumeister. [9] These authors jointly supervised this work: Marine Vallet, Georg Pohnert. *email: mvallet@ice.mpg.de; georg.pohnert@uni-jena.de

Omycetes are major pathogens of plants and animals[1–4]. These parasites also infect fish[5–7] and algae[8] in marine and freshwater ecosystems, thereby causing economic loss in fisheries and aquaculture[9,10]. Parasitic oomycetes developing inside the host cells occur frequently during microalgal blooms where they can control the plankton species succession during short-term epidemics. Little is known about the underlying molecular mechanisms of algal infection due to limitations in establishing a stable pathosystem for laboratory investigations[11–15]. In natural plankton communities, parasites shift from a flagellated free-living infectious stage (the zoospore) to an intracellular stage (the holocarpic thallus)[1]. Single-celled zoospores attach on the algal host surface and penetrate into the host cell where they develop a thallus. In the late infection stage, the entire thallus of the holocarpic pathogen is converted into a sporangium and zoospores are liberated through exit tubes[16]. This step is paramount to the completion of the life cycle, allowing the parasite dissemination to new algal cells. Signaling cues are involved in the chemotaxis of the flagellated parasitic forms[17], but the cellular effectors of oomycete infection of microalgae are unknown. The cultivation of the parasite and its host in an in vitro pathosystem can enable the investigation of infection mechanism, as demonstrated for the oomycete *Eurychasma dicksonii* that broadly infects seaweeds[15,18,19]. *Langenisma coscinosdisci* is a parasitoid infecting the marine genus *Coscinodiscus* that forms massive blooms in cold coastal waters of the Northern Hemisphere[16]. This parasite is often reported in field studies[20,21] and can be isolated in biphasic culture[22]. The role of *L. coscinodisci* in the breakdown of its host's blooms has yet to be determined.

Here, we use a stable laboratory pathosystem established by Buaya et al.[23] to investigate how the algal metabolome is rewired during the infection process. Two algal alkaloids that are substantially up-regulated during infection were identified, localized, and functionally characterized. These carbolines inhibit the algal growth and induce the cell plasmolysis in healthy *Coscinodiscus granii*. When added to infected cultures they also increase the infection rate. This hijacking of the host's alkaloid metabolism is unique in oomycete infections and proves a remarkably specialized strategy.

## Results

**Response of *C. granii* metabolome to oomycete infection.** Healthy and *Lagenisma*-infected cells of the centric diatom *C. granii* were isolated from Helgoland (North Sea) waters during a summer phytoplankton bloom[23]. The parasite *L. coscinodisci* can be maintained by re-inoculation of infected cells into healthy host cultures every four days. The oomycete produces short-lived biflagellate zoospores, which attach to the host's cell surface (Fig. 1a). These penetrate into the host and develop into stout filamentous hyphae growing within the algal cytoplasm. The parasitic thallus is then converted into a sporangium, in which new infectious zoospores maturate (Fig. 1a), before their release that occurs within seconds through an exit tube (Supplementary Movie 1).

Preliminary experiments showed that filtrates of infected cultures did not cause an induced resistance of *C. granii* to the parasite (Supplementary Table 1). Also, the growth of *C. granii* was not impacted upon exposure to filtrates of infected cultures (Supplementary Fig. 1). To further study the cellular metabolic response of *C. granii* to infection or exposure to exudates from infected cells, an untargeted metabolomics study was conducted with ultra-high performance liquid high-resolution mass spectrometry (UHPLC-HR-MS). Therefore healthy and infected cells were grown in a non-contact dual cultivation apparatus (Fig. 1b)[24]. This setup allowed

the exchange of exudates between treatments, whether healthy, infected, or exposed (that is, uninfected cells exposed to the exudates of a neighboring infected population) but separated the cells physically. Healthy cell controls reached the highest cell counts after 5 days of culturing, exposed cells were slightly reduced in growth, while few cells survived in the infected chambers (Supplementary Table 2). *C. granii* grown in the non-contact co-cultures were harvested, extracted, and analyzed in UHPLC-HR-MS. The analysis in positive polarity yielded more chemical information than in negative mode and was therefore initially selected for further investigation. Peak intensities were normalized to the cell density (Supplementary Table 2), yielding a matrix with 2601 signals (molecular formula, *m/z*, retention time). Metabolic profiles of extracts from healthy, exposed, and infected cells were clearly discriminated by principal component analysis (PCA) with a total principal component variance of 75.2% (Fig. 1c). Among the signals that are elevated in infected extracts, 21 metabolites were tentatively elucidated with high resolution MS/MS, of which 5 were further confirmed with authentic standards (Fig. 1d, Supplementary Table 3, Figs. 6–14). All identified metabolites are likely of algal origin since they were also detected in minor amounts in healthy diatoms and/or cells exposed to exudates of the infected cells. Among the most significantly up-regulated metabolites were β-carboline and 4-carboxy-2,3,4,9-tetrahydro-1H-β-carboline (4-CTC), which were identified after evaluation of their HR-MS and MS/MS spectra by co-injection with analytical standards (Supplementary Figs. 6 and 7). In addition, a number of highly abundant dipeptides (Supplementary Fig. 9) and lysophosphatidylcholines (Supplementary Figs. 10, 12 and 13) were detected in the extracts of infected cells as well as a putative dichloro-4-methylquinoline (Supplementary Fig. 14).

**Quantification and localization of carbolines in diatom cells.** Among the metabolites up-regulated during infection, the alkaloids β-carboline and 4-CTC were of interest due to their reported bioactivity[25–27]. The compounds were detected in fmol per cell amount, corresponding to up to 30 μM intracellular concentration in infected diatoms (Fig. 2a). These concentrations exceed those in uninfected cells more than 14-fold for 4-CTC and 56-fold for β-carboline, demonstrating the response of the diatoms towards infection. In accordance with the observed lack of an induced response, these compounds were not significantly elevated in cells exposed to the exudates of infected cells. Both compounds were also detected in the exudates of infected cells in concentrations reaching the pM range. Concentrations in healthy cell exudates were below the limit of quantification (Supplementary Fig. 5, Table 4). To assess if the single cells respond to the pathogen or if rather community responses are observed, atmospheric pressure matrix-assisted laser desorption/ionization high-resolution mass spectrometry (AP-MALDI-HR-MS) with a cellular resolution was performed. Therefore, infected and healthy cells that were selected by light microscopy were profiled. The single-cell mass spectra were processed and compared using our recently established workflow[28]. The signal intensity of β-carboline at *m/z* 169.0761 $[M + H]^+$ in all infected algal cells was significantly elevated compared to control cells (*t*-test, $n = 7$, *p* value < 0.0001) (Fig. 2a). Relative quantification was achieved by analyzing the standard deposited on cleaned and empty *C. granii* shells (Supplementary Fig. 2). The limit of detection and limit of quantification are 3.4 and 34.1 fg per diatom shell, respectively. To further determine the intracellular localization of carbolines we could exploit their fluorescent properties and conducted confocal laser scanning microscopy (cLSM) on single algal cells. As the β-carboline and 4-CTC both possess similar autofluorescence, the sum of their emissions could be investigated

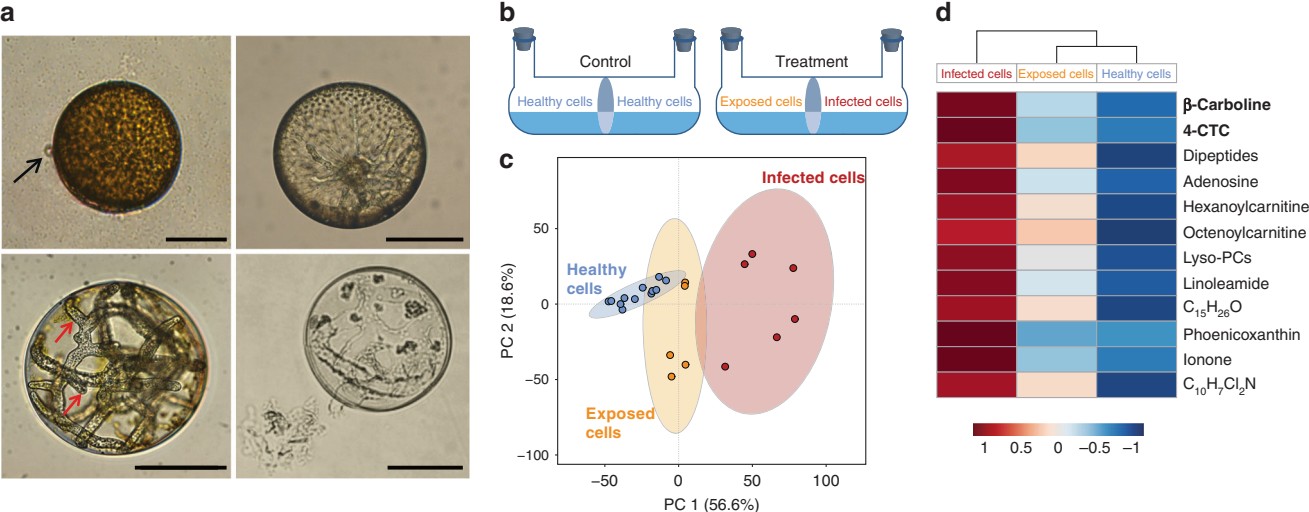

**Fig. 1** Metabolomics investigation of the oomycete infection in the marine diatom *Coscinodiscus granii*. **a** Life stages of the parasitoid oomycete *Lagenisma coscinodisci* infecting the diatom host *Coscinodiscus granii*. A single zoospore (highlighted by black arrow) attached to the algal surface (upper left). *L. coscinodisci* hyphae developing within the algal cytoplasm (upper right). The thallus converts into a sporangium from which mature infectious zoospores are produced (lower left). After zoospore release the empty parasitoid thallus remains in the shell of the dead diatom (lower right). Scale bars equal 50 μm. **b** Co-cultures of healthy or infected diatoms in chambers separated by a membrane that allows the passage of exudates between the physically separated cells. The controls consist of healthy cells in both compartments. In the treatment, healthy cell populations were exposed to the chemical cues released by infected algal cells. **c** The three treatments (healthy, exposed, or infected) can be discriminated after 5 days of co-culturing by their cellular metabolome in the PCA score plot. The ellipses represented the 95% confidence region predicted by the statistical calculation. **d** Cellular metabolites up-regulated during infection and exposure are represented in a heatmap based on their relative MS intensities (red color for high intensity, blue for low intensity)

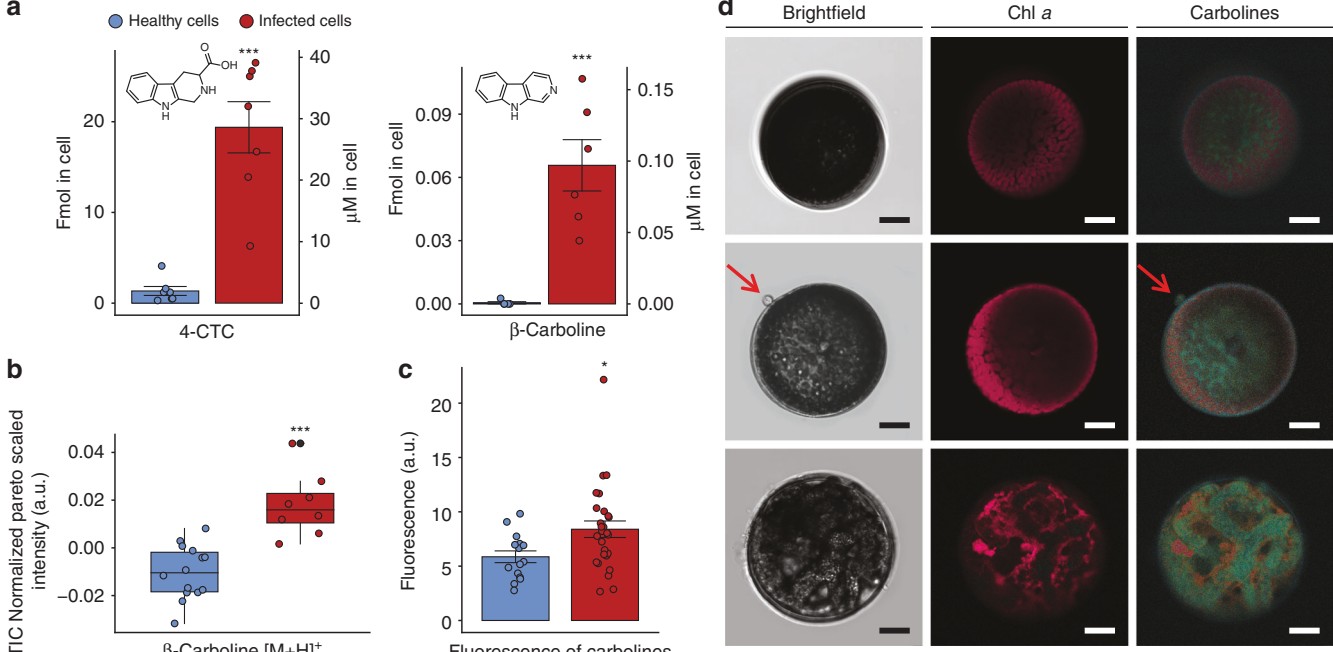

**Fig. 2** Quantification of carbolines in healthy and oomycete-infected diatom cells. **a** The carbolines quantified with UHPLC-HR-MS/MS are elevated in extracts from infected cells (left *Y*-axes). The intracellular concentrations in μM are calculated based on the cell volume (right *Y*-axes). The Student's test for both carboline concentrations found statistical significance between healthy and infected extract (Mann–Whitney rank sum test for 4-CTC, $n = 7$, $T = 77\,000$; Student's *t*-test for β-carboline $t = 4.9$, df $= 9$, *** for *p* value <0.001 for both tests). **b** Pareto scaled TIC normalized intensity of the β-carboline ion trace *m/z* 169.0761 was significantly higher in the AP-MALDI-HR-MS profiles of infected algal cells (Student's *t*-test, healthy cells $n = 14$, infected cells $n = 8$, $t = -5.3$ df $= 20$, *** for *p* value <0.001). **c** Means of cellular carboline fluorescence intensities (Mann–Whitney rank-sum test, $n = 15$ for healthy cells, $n = 27$ for infected cells, $T = 233\,000$, * for *p* value <0.05). All error bars indicate the standard error of mean. The box-plots are displayed with the maximum, minimum, median lines, and first/third quartiles and the samples as points. The outliers are displayed with a black dot. **d** The spatial localization and accumulation of the carbolines was observed with cLSM in healthy (upper row), early infected (middle), and late infected (lower row) cells. The microscopy pictures were taken in bright field (left) and the autofluorescence emission of chlorophyll *a* (Chl *a*) (middle) and carbolines (right) were recorded with an excitation wavelength of 405 nm. Scale bars equal 10 μm

under 405 nm excitation, yielding a maximal emission at 520 nm (Supplementary Fig. 4). The β-carboline/4-CTC fluorescence was detected in infected cells and increased drastically until reaching a maximal intensity in the late infection stage. Just before emergence of the zoospores, highest concentrations of the compounds were observed in the sporangium (Fig. 2c). The fluorescence intensity was significantly increased in infected cells averaged over all infection stages ($8.4 \pm 0.7$ a.u.), compared to healthy cells ($5.8 \pm 0.5$ a.u.) (Fig. 2c).

### β-Carboline and 4-CTC in algal growth regulation and infection.
To assess the role of the carbolines, a series of bioassays was performed in which healthy C. granii cells were treated with 4-CTC and β-carboline. Their growth was significantly inhibited at threshold doses of 24 μM (Fig. 3a). For 4-CTC this corresponds to the intracellular concentrations observed in infected cells (Fig. 2a). In addition, 4-CTC treatment increased the infection success of L. coscinodisci substantially even at 0.6 μM concentration (Fig. 3b). The compounds also triggered plasmolysis in healthy C. granii cells by inducing a vacuolization and cytoplasmic detachment (Fig. 3c). The cellular phenotype observed was remarkably similar of that of infected cells (Fig. 3c).

## Discussion
We used a pathosystem established from field samples comprising the pathogenic oomycete L. coscinodisci that infects bloom-forming algae of the diatom genus Coscinodiscus (Bacillariophyceae)[23]. The oomycete is an obligate and specialist parasite. After infecting a Coscinodiscus cell, it requires a minimum of 2 days to complete its life cycle, effectively killing the host. Most pathogenic oomycetes of plants, humans, and fish use host nutrients to develop and form sporangia[29], and given its exclusive intracellular development a similar strategy is likely for L. coscinodisci.

The metabolomics investigation of algal populations in different infection stages shed light on the regulation of metabolites during the development of the oomycete disease. The metabolic patterns of algae that were exposed to the chemical exudates of infected differed from those of healthy cells. Changes were even more pronounced in infected cells (Fig. 1c). Several cellular metabolites were significantly elevated in infected cells and further structurally elucidated. Tandem mass spectrometry enabled the identification of 21 metabolites up-regulated during infection (Fig. 1d, Supplementary Table 3, Figs. 7–14). These include dipeptides and nucleic acids, which point towards metabolic reprogramming by oomycete effectors[30,31]. In addition, lysophosphatidylcholines, linoleamide, hexanoylcarnitine, and octenoylcarnitine were increased in infected cells, which might enhance disease susceptibility[32] and reflect algal stress[33].

The PCA and heatmap (Fig. 1c, d) of the metabolomics data revealed a significant induction of two indole alkaloids arising from the tryptophane pathway, β-carboline and 4-CTC (Supplementary Figs 6, 7). β-Carboline is a widely distributed indole alkaloid that is also found in human tissue, scorpions' cuticle, medicinal plants, freshwater bacteria, and marine cyanobacteria[26,34]. This compound from industrial sources is monitored in aquatic ecotoxicology due to its activity against several cyanobacteria species[35]. AP-MALDI-HR-MS investigations proved that the elevated production of the carbolines was shared by all infected cells from the population while non-infected cells contained only minor amounts. This demonstrates that the induction of these cellular signals is a process triggered universally within infected cells. There is thus no substantial inter cellular variability that would explain survival of individual resistant cells within a population. The rather universal infection

of all cells could explain a bloom termination as consequence of a L. coscinodisci outbreak. The intracellular localization of the metabolites using cLSM revealed that the compounds are elevated in all infection stages but an increase in concentration is observed from zoospores attachment till sporangium formation. As the alkaloids co-localized with the oomycete cellular structures in the last infection stage (Fig. 2d), we assume that these compounds are taken up into the sporangium. In our functional assays, the carbolines inhibited algal growth in a dose-dependent manner (Fig. 3a) and promoted the infection when supplemented in the medium (Fig. 3b). The concentrations required to promote the processes were similar to those reached within infected cells hinting at a potential involvement as virulence factors. The carbolines also induced morphological alterations similar to those found in untreated infected cells at the active concentration (Fig. 3c). The phenotype was similar to a plasmolysis. In most studied cases of plant/oomycete interactions, the parasite triggers innate immune responses in the host, through elicitors such as branched β-1,3-glucans and eicosapolyenoic acids[36]. As a consequence, plants can produce defense metabolites to effectively kill the oomycete. It is a common strategy for plants to counteract predator and pathogen invasion by the induction of small molecules[30]. Indole secondary metabolites such as camalexin[37] but also oxylipins[38] take part in the regulation of plant responses to oomycete attack. A rewiring of the metabolome upon infection was also observed in C. granii. However, the up-regulated metabolites apparently do not contribute to resistance since cells exposed to lysed cultures contained intermediate concentrations of the up-regulated metabolites. The rewiring of the host metabolome in C. granii thus serves not an induced defense but rather the pathogen´s success. Such a strategy, where induced host metabolites increase infection was also observed during the viral infection of the coccolithophore microalga Emiliania huxleyi[39]. There the virus triggers production of algal sterols that promote virulence. Our results provide first insights into the molecular process of the specialist oomycete L. coscinodisci that infects and kills the bloom-forming diatom C. granii. We show the effects of two indole alkaloids belonging to the class of β-carbolines, and show that the metabolites of healthy and infected diatom cells differ markedly, demonstrating that the oomycete L. coscinodisci manipulates the algal metabolome during the completion of its life cycle.

## Methods
**Purchased standards.** Analytical standards (4-CTC, adenosine and lysophosphatidylcholine) were obtained from Sigma-Aldrich, β-carboline from Acros Organics, linoleamide from Santa Cruz Biotechnology, chlorophyll a from Wako. Internal standards for UHPLC-HR-MS analysis were p-fluoro-L-phenylalanine, p-fluorobenzoic acid, and decanoic-d$_{19}$ acid all obtained from VWR.

**Microbial strains.** Wild host species C. granii and the parasitoid oomycete L. coscinodisci strain LagC7 were collected and purified as described in Buaya et al.[23] Briefly, plankton samples were collected at Helgoland Roads using plankton nets. Single diatom cells were picked and transferred into sterile Guillard's (f/2) enrichment medium (Sigma-Aldrich) prepared with natural seawater (ATI). Cultures were maintained under fluorescent lamps (irradiance 100 mE m$^{-2}$ s$^{-1}$) with a 14 h photoperiod coupled to a thermoregulated cycle (16−12 °C day–night). While single-cell isolates of the host diatoms were established, L. coscinodisci was maintained on mixed diatom samples, cultivated like the single diatom species cultures, but in 9 cm Petri dishes (Sarstedt). After 3–5 days, single diatom cells of C. granii were taken from the individual diatom cultures and transferred three times through sterile seawater using a 100 μL pipette and then infected by co-cultivation with infected cells. When the pathogen thalli became apparent, diatoms with a single thallus were again transferred to an uninfected batch of the host. Microscopy was performed using an inverted microscope (AE30, Motic) for regular checks and sub-cultivation, and a Zeiss Imager 2 (Carl Zeiss) to record microscopic images[22]. The infection was maintained by inoculation of exponentially growing 30 mL host cultures with a 20 μL aliquot of infected cells every 3 to 5 days. The C. granii and L. coscinodisci strains are currently maintained in vitro and available upon request from Marco Thines (m.thines@thines-lab.eu). C. granii infected by the Lagenisma

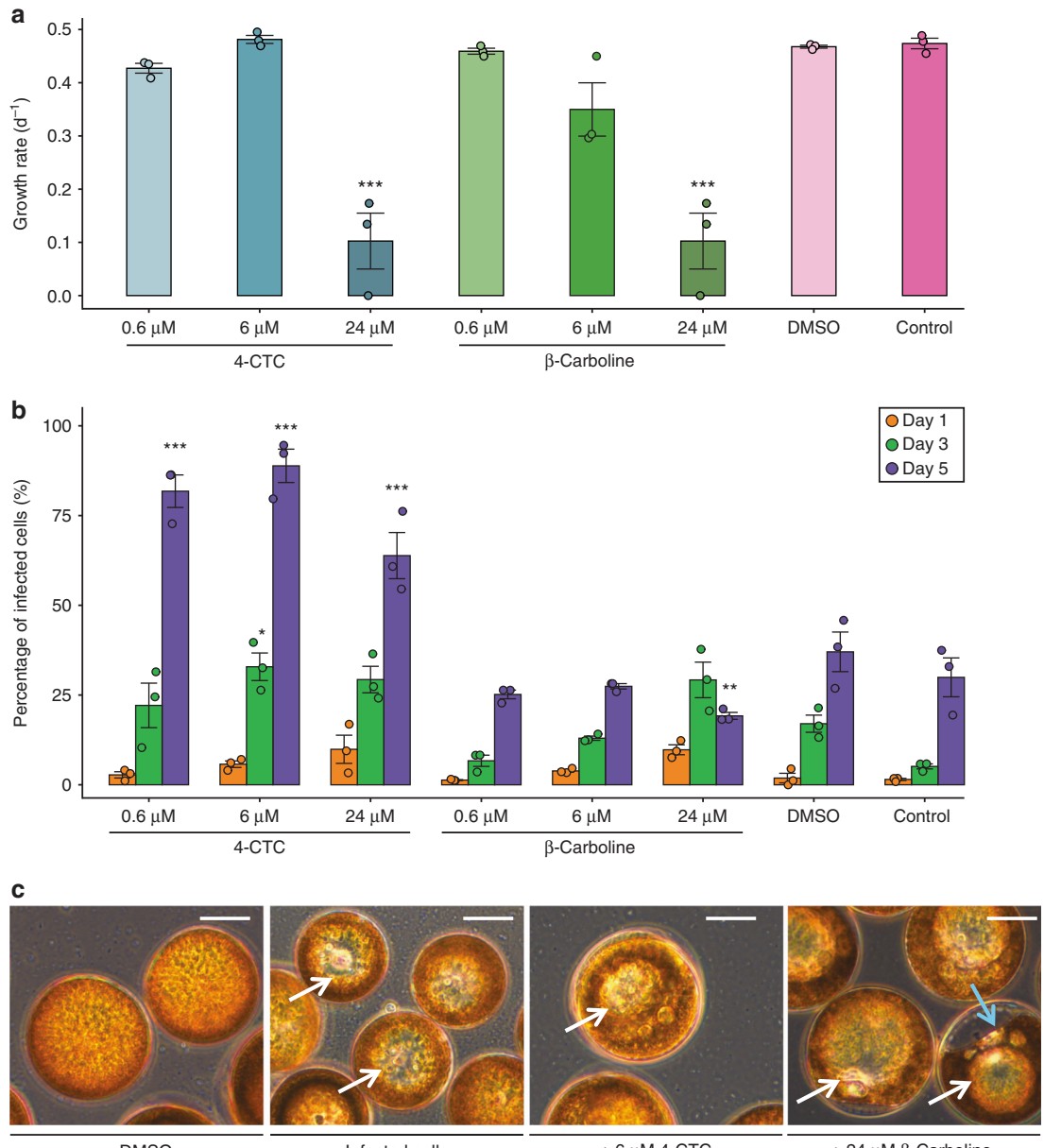

**Fig. 3** Functional assays showed that the carbolines impact the growth, morphology, and infection rate of the marine diatom *C. granii*. **a** β-Carboline and 4-CTC inhibit the growth of *C. granii* algal cells at 6 and 24 μM, respectively (mean growth rates ± SE, one-way ANOVA with multiple comparisons against the DMSO group, Holm–Sidak method, $F = 26$, $dF = 8$, $n = 3$, *** for $p$ value >0.001). **b** The infection rate was elevated in cultures treated with 4-CTC (>6 μM) after 3 and 5 days of incubation. Statistical tests, a two-way repeated measures ANOVA with pairwise multiple comparison ($F = 17$, $dF = 8$, $n = 27$) and the Dunnett's method with multiple comparisons against the DMSO group found significant differences with the treatments highlighted by *** for $p$-value <0.001, ** for $p$ value <0.01, * for $p$ value <0.05. Mean of the infection rates was determined ±SE. **c** The phenotype of cells treated with β-carboline at 24 μM and 4-CTC (6 μM) (pictures on the right) was similar to the untreated infected cells (second picture from left). Plasmolysis was observed with vacuolization (white arrows) and some cytoplasmic detachment (blue arrow). Scale bars equal 20 μm, representative pictures of three replicates are shown

*coscinodisci* strain LagC7 is available as freeze-dried sample and the specimen was deposited in the Senckenberg Museum of Natural History, Cryptogams Section, Frankfurt am Main, under the number FR-0046113.

**Infection experiments**. To assess the effect of the exometabolome of infected algal cultures on the growth and resistance of *C. granii*, 5-day-old-infected *C. granii* cultures (200 mL) were sterile filtrated through a 0.22 μm Steritop® vacuum filtration system (Merck Millipore) and the spent medium was used for inoculation as follows. The cell density of exponentially growing healthy culture was determined and the cells were harvested by pouring the culture in 40 μm pore-size nylon mesh (Corning Life Sciences), washed with 20 mL sterile seawater, inoculated in 30 mL spent medium at a final concentration of 10 cells mL$^{-1}$. Growth was monitored over 8 days. To verify the infection rate after 5 days, three samples of treated cells and controls were infected with 2.6 mL of an LagC7 infected culture. The infection

rate was followed over 4 days by light microscopy. Cell densities were counted in a Sedgewick Rafter Counting Chamber (Pyser-SGI).

**Co-culture experiments**. Co-culturing chambers consisting of two separate vessels separated by a 0.22 μm pore-size membrane sterile filter (Millipore) were built and assembled as described in detail in Paul et al.[24] but with a scaled-down total volume of 240 mL. The experiments were conducted in six biological triplicates. For inoculation, exponentially growing *C. granii* were harvested by pouring the cultures (200 mL) in 40 μm pore-size nylon mesh (Corning Life Sciences). Cells were washed with 20 mL sterile seawater and transferred by pipetting to the co-culturing set-up filled with 120 mL sterile f/2 medium. Both chamber compartments were inoculated using this protocol. For infection, with *L. coscinodisci* strain LagC7, 2.6 mL of an infected culture were added in the respective compartment. Cells were counted after 5 days of incubation under culturing conditions described above and

harvested for metabolomics. Culture medium was incubated as above-mentioned to serve as blank.

**Untargeted metabolomics profiling of cellular extracts.** Cultures were filtered under vacuum on 25 mm GF/C microfiber filters (Whatman) disposed on filter plates in a filtration unit (VWR International). The GF/C filters were directly transferred to 2 mL safe-lock Eppendorf tubes and extracted with 1.5 mL methanol (Sigma-Aldrich) under sonication in an ultrasound bath for 2 min. Supernatants were transferred to new tubes and centrifuged for 20 min at 12 000 g. Organic phases were transferred into new glass vials and dried in a Vacufuge® plus vacuum concentrator (Eppendorf). The extracts were taken up with 50 μL methanol and 5 μL of each sample were transferred into a QC mix sample. Internal standard mix (10 μg mL$^{-1}$) was added and 10 μL were injected into the UHPLC-HR-MS (Ulti-Mate 3000 UHPLC Dionex) connected to an Accucore C-18 column ($100 \times 2.1$ mm, 2.6 μm) coupled to a Q-Exactive Plus Orbitrap mass spectrometer (Thermo Fisher Scientific). The metabolite separation was performed in a 14 min gradient, starting with 100% of aqueous phase (2% acetonitrile, 0.1% formic acid in water) and increasing with the acetonitrile phase (0.1% formic acid in acetonitrile) within 9 min until reaching 100%. This was held for 4 min before switching back to 100% of water phase and equilibration for 1 min. Throughout the rising gradient the flow rate was gradually increased from 0.4 to 0.7 mL min$^{-1}$ in 9 min, and set back to 0.4 mL min$^{-1}$ afterwards. Mass spectrometry was conducted in positive-ion mode with a scan range of $m/z$ 100 to 1500 at a peak resolution of 280 000. AGC target was set to $3 \times 10^6$ and maximum ion time was set to 200 ms. The MS/MS spectra of precursor ions, selected with an inclusion list, was obtained from cell extracts with the above-mentioned parameters, and within an isolation window of $m/z$ 0.4 at a peak resolution of 280 000 (NCE 15, 35, 45). The raw dataset was uploaded on MetaboLights [https://www.ebi.ac.uk/metabolights/MTBLS775].

**Analysis of significant metabolites and in silico identification.** Raw data were imported into Compound Discoverer$^{TM}$ software 2.1 (Thermo Fisher Scientific) for peak picking, deconvolution, and identification of the metabolites. Mass tolerance for MS identification was 5 ppm, minimum MS peak intensity was $2 \times 10^4$, and intensity tolerance for isotope search was 50%. Relative standard deviation value was set to 50%. The injection sequence of the samples, cell density, and normalization factors are listed in Supplementary Table 2. The compound list was exported as.csv file, the intensities were normalized based on cell density count and analyzed with MetaboAnalyst 3.0[40]. PCA was performed to compare metabolites similarities between cellular extracts. The identity of selected ions was confirmed with tandem mass spectrometry and the MS/MS spectra (Supplementary Figs. 7–14) were compared by spectral similarity search in PubChem and with those of analytical standards. Assignment of putative identities of selected ions was performed through database search of selected MS$^2$ spectra in PubChem using CSI:FingerID[41]. If possible, the actual identity was confirmed by comparison of retention time and MS$^2$ spectra of an analytical standard.

**Absolute quantification of the carbolines.** Independent cultures were used for absolute quantification in infected and healthy cells. The cell volume was calculated based on the average measurements of 12 C. granii cells using a light microscope and Image J 1.52a[42]. Extraction and UHPLC-HR-MS analyses were performed as mentioned above. Extracts (5 μL) were injected and the mass spectrometer was run in a Parallel Reaction Monitoring (PRM) mode. Separation started with 100% of 2% acetonitrile, 0.1% formic acid in water at a flow rate of 0.4 mL min$^{-1}$ and a linear gradient to 100%, 0.1% formic acid in acetonitrile after 9 min at 0.7 mL min$^{-1}$ was programmed. The AGC target was set to $2 \times 10^5$, the maximum ion time to 200 ms. Precursors were selected within an isolation window of $m/z$ 0.4 at a resolution of 35 000. Selected transitions for β-carboline were $m/z$ 169.0760 → 115.0536, 142.0644 (NCE 100), and 4-CTC $m/z$ 217.0972 → 144.0808 (NCE 40). One microliter of the standard solution (4-CTC 1–1000 ng mL$^{-1}$, β-carboline 0.1–3.5 ng mL$^{-1}$) were injected and the raw data were analyzed with QuanBrowser of Thermo Xcalibur 3.0.63. The calibration curves were determined for each standard (Supplementary Fig. 3). The concentrations calculated for β-carboline were set to 0 when the intensities were below the range determined by the calibration done with the concentration of the standard. To quantify the amount of both compounds in the exudates of algal cells (healthy or 3-day-old infected cells), 100 mL of cultures were sterile filtrated with a 0.22 μm membrane and extracted with solid phase extraction cartridges (6 cc OASIS HLB sorbent, Waters) with a 10 mL water-washing step and 3 mL of 100% methanol for the elution. The analytical standards prepared in methanol solution were added to 100 mL of sterile f/2 nutrients medium (final concentration β-carboline 1–64 pM and 4-CTC 1–50 pM) in independent triplicates and the media were extracted as described above to determine the calibration curves (Supplementary Fig. 5, Table 4). Limits of detection and quantification were determined according to Reichenbächer et al.[43] After drying the extracts under a continuous nitrogen flow, 200 μL of 70% methanol in water were added and 10 μL were injected for UHPLC-MS/MS analysis in the conditions for MS/MS quantification of the carbolines as above-mentioned. The calibration curves were determined for each standard (Supplementary Fig. 5).

**Single-cell studies with AP-MALDI-HR-MS and cLSM.** Single diatom cells in healthy or infected state ($n = 7$) were profiled in positive ionization mode with an AP-SMALDI10 MALDI-HR-MS (TransMit). Cells were analyzed in positive polarity with the number of laser shots per spot set to 30 (approximately 1.5 μJ × shot$^{-1}$) within the laser frequency of 60 Hz. Mass spectra were recorded in the mass range from $m/z$ 100–1000 with the peak resolution of 280 000[28]. The detection of the carbolines was achieved by applying an aqueous 2,5-dihydroxybenzoic acid (DHB) matrix solution (20 mg mL$^{-1}$) to cells deposited onto a wetted GF/C filter (10 mm$^2$). Methanolic solutions of each standard (0.2 mg mL$^{-1}$) were mixed in an 1:1 (v:v) ratio with the DHB solution and 0.1 μL were applied on HTC printed microscope glass slides (Omni Slide Hydrophobic Array $26 \times 76$ mm, 66 well; Prosolia Inc.). The β-carboline was detected at $m/z$ 169.0758 as [M + H]$^+$ (Supplementary Fig. 2b) and the intensities were recovered, TIC normalized, and Pareto-scaled, following the script from Baumeister et al.[28] To determine the limit of detection and limit of quantification of β-carboline, C. granii shells were obtained by incubating cells in 80% H$_2$O$_2$ for 2 h at 80 °C. Six microliters of diatom shells (methanolic suspension 0.3 mg mL$^{-1}$) spotted on PTFE slides were covered with the standard in methanol solution (1 μg mL$^{-1}$ to 1 ng mL$^{-1}$), air-dried and 1 μL of DHB matrix (20 mg mL$^{-1}$ in methanol) was deposited on the whole area (Supplementary Fig. S2). LDI-MS profiling was performed by analyzing single shell ($n = 26$) for each concentration. The means and standard deviation of the total surface area and area per shell were determined from 10 spots and 21 shells, enabling the quantification of β-carboline per shell and determination of LOD and LOQ, following recommendations from the European Union regulation 2002/657/EC[44]. Total surface area and shells areas were determined from pictures obtained with digital microscope Keyence VHX-5000. The acquired data are available in the Source data file. Confocal laser scanning microscopy was conducted on standards and cell samples to detect the carbolines presence in single cell. The carbolines (10 mg mL$^{-1}$) and chlorophyll a (5 mg mL$^{-1}$) were prepared in DMSO (Roth) and 20 μL were spotted on the microscope slide. After an excitation at 405 nm, β-carboline and 4-CTC emitted a broad autofluorescence emission ranging from 450 to 600 nm, with peak maximum at 520 nm. The autofluorescence spectra were acquired in lambda mode (Supplementary Fig. 4). To separate the strong chlorophyll a autofluorescence and retain a clear display of the β-carboline, the lambda detection was cut at 633 nm. The cells were scanned separately and the chlorophyll a autofluorescence signal was occluded in the carbolines scans to get a clearer signal. Pictures were captured on a confocal laser scanning microscope 880 with a 20×/0.8 Plan-Apochromat objective (Zeiss).

**Functional bioassays.** The growth of diatom C. granii was monitored in presence of the β-carboline and 4-CTC (final tested concentrations ranged from 0.6 to 60 μM) over 12 days incubation in six-well plates (7 mL f/2 medium, 10 cells mL$^{-1}$ initial concentration). Controls consisted of treating the cells with DMSO 0.1% (Roth) or without treatment. Cell density was measured after 12 days ($t$) and the growth rate $\mu$ was derived from: $\mu = t^{-1} \times \ln(N_t/N_0)$ in day$^{-1}$ where $N_0$ is the initial cell count and $N_t$ the cell count after the incubation time[45]. Diatom cells were cultivated in six-well plates (10 mL f/2 medium ± treatment as above-mentioned) were infected with a single cell and the sporangia number was counter every 2 days for 5 days. The infection rate was derived from (number of sporangia/total number of cell) × 100%. All experiments were conducted in biological triplicates. Pictures of cells were obtained from an Axiovert 200 microscope with a 20×/0.4 Ph2-Korr-Achroplan objective (Zeiss). The statistical tests were performed with SigmaPlot 12.0 (Systat Software).

**Reporting Summary.** Further information on research design is available in the Nature Research Reporting Summary linked to this article.

## Data availability
The metabolomic datasets generated and analyzed during the current study are available in the Metabolights repository [https://www.ebi.ac.uk/metabolights/MTBLS775]. All living strains are available upon request to Prof. Marco Thines (m.thines@thines-lab.eu) and a fixated specimen of the pathosystem C. granii/Lagenisma is deposited in the Senckenberg Museum of Natural History, Cryptogams Section, Frankfurt am Main, under the number FR-0046113. The data underlying Figs. 2, 3 and Supplementary Figs. 1, 2, 3, 4, 5 and Table 1 are provided as a Source Data file.

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

## Acknowledgements

This work was supported by a Fellowship from the Max Planck Society awarded to G.P. and the German Research Foundation (DFG) CRC 1127 "ChemBioSys". This work was also funded by the DFG under Germany´s excellence strategy EXC 2051 Project ID 390713860. A.B. is supported by the Katholischer Akademischer Ausländerdienst (KAAD) through a Ph.D. fellowship. M.T. is supported by LOEWE in the framework of the Center for Translational Biodiversity Genomics (TBG) funded by the government of Hessen. We are grateful to Alexandra Kraberg, Sebastian Ploch, and Manuel Arnold for laboratory support.

## Author contribution

A.B. and M.T. isolated, identified, and provided the algal and parasitic strains. M.V. and T.U.H.B. developed the metabolomics pipeline and analyzed the data. V.G. acquired and analyzed the fluorescence microscopy data. F.K. performed the AP-MALDI-HR-MS analysis. M.V. analyzed the MS/MS data with contributions from T.U.H.B. and A.S. M.V. and G.P. conceived the study, directed all experiments, and wrote the manuscript with contributions from the co-authors. All authors approved the manuscript.

## Competing interests

The authors declare no competing interests.
