## [Peer Review File · Nature Communications]

Reviewers' comments:

Reviewer #1 (Remarks to the Author):

This is an interesting study. Authors utilized the LCMS metabolomics and in situ mass spectrometry to reveal the metabolites involving the interaction between *Lagenisma coscinodisci* and *Coscinodiscus granii*. However, there are some concerns about the experiments and data interpretation.

1. Figure 1a, the black arrow is missing in the upper left picture.
2. In Figure 1b, the membrane supposed to prevent the infection and direct contact from exposed cells and infected cells. Lines 99-100, "Healthy *C. granii* cells that were exposed to the filtrates of infected cultures could still be infected", what do you want to demonstrate about the "could still be infected"?
3. In Figure S7, the identification of farnesol is incorrect because the retention time of the one from infected cell extract (6.91 min) is very different from the standard (7.62 min). The MS/MS fragments are also different since the same HCD (31.67) was applied to both samples. The RT recorded in Figure S7 was also different from that in Table 1 (7.2 min).
4. Table 1 can be moved to SI.
5. The untargeted metabolomics profiling data were only collected using positive mode. The metabolites which are only ionized in negative mode may also contribute to the important biological functions in the co-culture condition.
6. The quantification results of β -carboline and 4-CTC using LCMS, AP-MALDI-MS, and confocal laser scanning microscopy are very different, although they did show the same trend: the infected cells accumulated more β -carboline and 4-CTC than healthy cells. I understand authors did not provide the quantification data of 4-CTC using AP-MALDI MS because of the low intensity (Figure S2a). However, I cannot see the signal of β -carboline (m/z 169.07) in the spectrum of the infected cell, either (Figure S2d, only 171.0993 observed). Besides, in Figure 2b, the normalized intensity of β -carboline in the healthy cells is negative, why? What kind of normalization method was applied here? What are the LOD and LOQ of β -carboline and 4-CTC using AP-MALDI-MS? I don't see the useful information deduced from AP-MALDI-MS data in this study since the sensitivity and quantification are not promising as LCMS and no spatial resolution (such as imaging mass spectrometry or confocal) was provided here.
7. The title of Figure S2, "LDI-HR-MS" should be revised as "MALDI-HR-MS". You did not use the matrix-free approach in this study (The detection of the carbolines was achieved by applying an aqueous 2,5-dihydroxybenzoic acid (DHB) matrix solution ($20 \text{ mg} \times \text{mL}^{-1}$) to cells deposited onto a wetted GF/C filter ($10 \times 10 \text{ mm}$), lines 347-349).
8. In Figure 2a, the intracellular concentration of β -carboline is around $0.10 \mu\text{M}$, which is much lower than the threshold dose ($6 \mu\text{M}$) shown in Figure 3a. In addition, the effect of β -carboline on the infection success is not significantly different in comparing with DMSO treatment, except for the day 1 and day 3 data of $24 \mu\text{M}$ treatment, which is much higher than the intracellular concentration of β -carboline. The phenotypic data about 4-CTC treatment were missing, although 4-CTC showed the more promising effect on the infection success and the dose is close to the intracellular concentration. In addition, in Figure 3b, $0.6 \mu\text{M}$ of 4-CTC can significantly increase the infection rate at day 5. However, there is no significantly dose-dependent manner observed either in β -carboline or 4-CTC treatment to the infection rate. It is quite strange authors made the conclusion in lines 173-180 and Figure 3 based on such incomplete data.

9. The quantification observed in this study is the intracellular concentration of β -carboline and 4-CTC. However, the data shown in Figure 3 are the effects of exogenous β -carboline and 4-CTC to cell growth and morphologies. Interestingly, the data of Table S2 showed the infection rate at day 4 is similar between control and filtrate treatment. Do you have any explanation? Authors may consider quantifying the secreted concentration of β -carboline and 4-CTC in the co-culture condition.

Reviewer #2 (Remarks to the Author):

We know little about the chemical interactions between algae and eukaryotic pathogens. This concerns both micro- and macroalgae. On this front, the paper by Vallet et al. offers fascinating, significant new insight.

The approach and experimental work are rigorous. The authors convincingly show that infection of the diatom *Coscinodiscus* by the oomycete *Lagenisma* results in an upregulation of algal-synthesized alkaloids, and that these are accumulated by the parasite and facilitate its growth inside infected host cells. The paper is very well written, and images are of a high quality.

My only slight concern relates to the origin, accessibility and reproducibility of the biological material: would it be possible to deposit both a culture of the host and of the host with the pathogen, respectively, in a publicly accredited culture collection (such as RCC, CCMP or CCAP)? I am all too aware that maintaining such co-cultures over prolonged periods is challenging - I would therefore recommend looking into depositing cryoconserved cultures. At the very least, and if neither of these 2 options is feasible, I would request depositing fixed or freeze-dried material with PCR-friendly DNA in a herbarium such as PC, BM or UC.

Another minor point: I cannot see the black arrow in Fig. 1 a.

Overall, I think that this paper is well up to the standards of Nature Communications, and I recommend accepting it after minor revisions.

Reviewer #3 (Remarks to the Author):

This is a very interesting paper demonstrating that *Lagenisma* infection of a diatom induces the production of two indole alkaloids β -carbolines, and show that the metabolites of healthy and infected diatom cells differ significantly. From the data it is clear that the pathogen is manipulating the algal metabolome during the completion of its life cycle. These findings are not totally unexpected and one could make a pretty robust prediction that this would be the case, but it is the first time that it has been demonstrated for an oomycete host interaction, which is nice. What is unclear from the paper is how the two compounds actively promote the infection? Does it help the pathogen itself? Is it an essential compound that is needed for its growth and propagation or does an increased production inhibit a possible defence response in the host? Does it induce particular genes (possibly involved in suppressing host defence) being up regulated or maybe the compounds are downregulating host defence genes? Does it promote the production of particular nutrients for the pathogen? Such experiments would be relatively easy to do with the model infection system that the authors have set up. It requires the treatment of the diatoms with the two isolated compounds and a couple of controls and time points and perform a simple RNAseq experiment on

the cells treated. The results would be very informative as it will give you a very clear idea what genes may be affected and what pathways are triggered that may lead to a successful infection. Since the pathogen cannot be cultured in vitro without the host (my assumption), it will be more complicated to investigate what effect the compounds might have on the pathogen's own expression profile, but it might be possible to add extra β -carboline and 4-carboxy-2,3,4,9-tetrahydro-1H- β -carboline to an infected culture and see which genes are up-down regulated in the pathogen (and host). But I appreciate that this latter experiment could be a bit messy. I think to get a better picture about the role of these compounds is in the infection process, a simple RNAseq experiment, as suggested above, would greatly enhance the paper as it now reads as an unfinished (very nice) story.

Figure 1 does not have a black arrow to indicate the zoospore

Dot missing in line 166.

Line 213. Significant "t"

Line numbers in the replies refer to the manuscript with tracked changes.

Reviewer #1 (Remarks to the Author):

This is an interesting study. Authors utilized the LCMS metabolomics and in situ mass spectrometry to reveal the metabolites involving the interaction between *Lagenisma coscinodisci* and *Coscinodiscus granii*. However, there are some concerns about the experiments and data interpretation.

1. Figure 1a, the black arrow is missing in the upper left picture.

#The arrow has been added to Figure 1a.

2. In Figure 1b, the membrane supposed to prevent the infection and direct contact from exposed cells and infected cells. Lines 99-100, “Healthy *C. granii* cells that were exposed to the filtrates of infected cultures could still be infected”, what do you want to demonstrate about the “could still be infected”?

#We re-arranged the text that explains preliminary experiments (shown in the supporting Table S1 and Fig. S1). These were conducted to investigate a possible induced resistance of the alga to the pathogen (line 99ff). We then describe results from Fig. 1b in one block to avoid confusion. The second mentioning of induction (line 129) has been deleted.

3. In Figure S7, the identification of farnesol is incorrect because the retention time of the one from infected cell extract (6.91 min) is very different from the standard (7.62 min). The MS/MS fragments are also different since the same HCD (31.67) was applied to both samples. The RT recorded in Figure S7 was also different from that in Table 1 (7.2 min).

#We thank the referee for this observation. After additional co-injection with other farnesol isomers we can exclude that the metabolite is a farnesol isomer. We therefore changed the annotation of this metabolite to a “terpenoid” with identification MS3 level. The text has been modified accordingly. The change does not affect our conclusion since this metabolite was not further discussed.

4. Table 1 can be moved to SI.

#Table 1 has been moved to supplementary as Table S3.

5. The untargeted metabolomics profiling data were only collected using positive mode. The metabolites which are only ionized in negative mode may also contribute to the important biological functions in the co-culture condition.

#The analysis in positive polarity yielded more chemical information than the negative polarity and was hence further investigated. A sentence has been added line 112 to clarify. Since we could already identify the chemical mediators from this data set, we did not further pursue the

negative ionization data.

6. The quantification results of β -carboline and 4-CTC using LCMS, AP-MALDI-MS, and confocal laser scanning microscopy are very different, although they did show the same trend: the infected cells accumulated more β -carboline and 4-CTC than healthy cells. I understand authors did not provide the quantification data of 4-CTC using AP-MALDI MS because of the low intensity (Figure S2a). However, I cannot see the signal of β -carboline (m/z 169.07) in the spectrum of the infected cell, either (Figure S2d, only 171.0993 observed). Besides, in Figure 2b, the normalized intensity of β -carboline in the healthy cells is negative, why? What kind of normalization method was applied here? What are the LOD and LOQ of β -carboline and 4-CTC using AP-MALDI-MS? I don't see the useful information deduced from AP-MALDI-MS data in this study since the sensitivity and quantification are not promising as LCMS and no spatial resolution (such as imaging mass spectrometry or confocal) was provided here.

The entire experimental for this experiment has been expanded. Total Ion Current (TIC) normalization was applied to the intensity (as it is standard in the field of MALDI-MS). This information has been added in the material and methods (lines 379). The negative value arises from Pareto scaling – we include this information now in the figure legend.

Relative quantification was performed by applying the β -carboline (in methanol) on clean diatom shells and further identical treatment compared to the single cell MS. This information is now included in the manuscript (line 156 supporting material Fig S2). We refer to lines 378 to 389 for experimental details.

To show the β -carboline signal we now provide an insert with the zoomed region in the m/z range of the metabolite and its fragments in the MS profile of infected cells in Figure S2.

Signals for β -carboline (m/z 169.0760) and its fragments (m/z 142.0651, 115.0540) were detected with a detection limit of 1 fg per diatom shell. This is now documented in the new Supplementary Figure S2. The 4-CTC could not be detected neither in shells nor in single cells. It will not be feasible to do absolute quantification with live single-cell mass spectrometry as the matrix effect cannot be fully simulated — indeed the alkaloids are poorly soluble in aqueous solution and their analysis on clean diatom biomineralized shells would not consider the ionization-enhancement that might be observed when working with living cells. However, the relative quantification is fully valid and can, in the future, be used to document cellular individuality in the metabolic profile. Given the break-through that the response to infection can be shown for the first time in the MS of a single phytoplankton cell we would like to keep this result in the revised manuscript - now supported by data in Fig. S2, the experimental (line 372ff) and the deposited primary data.

7. The title of Figure S2, “LDI-HR-MS” should be revised as “MALDI-HR-MS”. You did not use the matrix-free approach in this study (The detection of the carbolines was achieved by applying an aqueous 2,5-dihydroxybenzoic acid (DHB) matrix solution (20 mg \times mL⁻¹) to cells deposited onto a wetted GF/C filter (10 \times 10 mm), lines 347-349).

#The title of Figure S2 has been modified accordingly

8. In Figure 2a, the intracellular concentration of β -carboline is around 0.10 μM , which is much lower than the threshold dose (6 μM) shown in Figure 3a. In addition, the effect of β -carboline on the infection success is not significantly different in comparing with DMSO treatment, except for the day 1 and day 3 data of 24 μM treatment, which is much higher than the intracellular concentration of β -carboline. The phenotypic data about 4-CTC treatment were missing, although 4-CTC showed the more promising effect on the infection success and the dose is close to the intracellular concentration. In addition, in Figure 3b, 0.6 μM of 4-CTC can significantly increase the infection rate at day 5. However, there is no significantly dose-dependent manner observed either in β -carboline or 4-CTC treatment to the infection rate. It is quite strange authors made the conclusion in lines 173-180 and Figure 3 based on such incomplete data.

#There might have been a misunderstanding caused by the different order of presentation in Fig. 2 and Fig. 3. We now adjusted the panel 2a to match the order of the results presented for each compound with that in panel 3a. It becomes obvious that 4-CTC concentrations in the infected cells are in the range required in the assays to observe activity. Beta-carbolide in contrast is active only in higher doses. In the text we are careful to discuss the activity of 4-CTC, beta-carbolide is included because of its biosynthetic relation and because we cannot rule out that external addition of the compound will not reflect the situation of intracellular production of the compound. The discussion is now made clearer not to make an incorrect claim.

We now added the phenotypic data about 4-CTC treatment (6 μM) in Fig. 3c. It shows that 4CTC impacts the algal cell morphology even in concentrations below than those observed in the cells.

We also adjusted Fig. 3b that was not supposed to show a concentration dependence but to verify the effects of the three treatments (compare Fig. 3a). We also delete the suggestion of a concentration dependence.

9. The quantification observed in this study is the intracellular concentration of β -carboline and 4-CTC. However, the data shown in Figure 3 are the effects of exogenous β -carboline and 4-CTC to cell growth and morphologies. Interestingly, the data of Table S2 showed the infection rate at day 4 is similar between control and filtrate treatment. Do you have any explanation? Authors may consider quantifying the secreted concentration of β -carboline and 4-CTC in the co-culture condition.

#We took into consideration the suggestion of the reviewer to investigate why the infection is similar between control and filtrate treatment. We performed solid-phase extraction of the exudates from healthy and infected algal cultures to assess the potential quantity of exogeneous carbolines in the filtrates, and also obtained calibration curves for the standards added to medium. We found that the carbolines are present in the pM range in the exudates of infected cells, and even below the limit of quantification for healthy cultures. These results were added to the manuscript (lines 139ff, 356ff and Supplementary Table S4, Figure S5). This explains why there was no effect on the infection rate when treating cells with filtrate (now table S1).

Reviewer #2 (Remarks to the Author):

We know little about the chemical interactions between algae and eukaryotic pathogens. This concerns both micro- and macroalgae. On this front, the paper by Vallet et al. offers fascinating, significant new insight.

The approach and experimental work are rigorous. The authors convincingly show that infection of the diatom *Coscinodiscus* by the oomycete *Lagenisma* results in an upregulation of algal-synthesized alkaloids, and that these are accumulated by the parasite and facilitate its growth inside infected host cells. The paper is very well written, and images are of a high quality.

My only slight concern relates to the origin, accessibility and reproducibility of the biological material: would it be possible to deposit both a culture of the host and of the host with the pathogen, respectively, in a publicly accredited culture collection (such as RCC, CCMP or CCAP)? I am all too aware that maintaining such co-cultures over prolonged periods is challenging - I would therefore recommend looking into depositing cryoconserved cultures. At the very least, and if neither of these 2 options is feasible, I would request depositing fixed or freeze-dried material with PCR-friendly DNA in a herbarium such as PC, BM or UC.

#In fact, the host / pathogen system can only be maintained with constant care. Therefore, the *Lagenisma coscinodisci* strain used in this study with its host *C. granii* were fixed and deposited in the herbarium collection of the internationally recognized herbarium of the Senckenberg Museum of Natural History, Cryptogams Section, Frankfurt am Main. This is now mentioned in line 279 and in the data availability statement line 430 the accession number is given. Strains will be made available on request (now indicated with the email address for enquiries in line 278

Another minor point: I cannot see the black arrow in Fig. 1 a.

#the figure has been modified accordingly

Overall, I think that this paper is well up to the standards of Nature Communications, and I recommend accepting it after minor revisions.

Reviewer #3 (Remarks to the Author):

This is a very interesting paper demonstrating that *Lagenisma* infection of a diatom induces the production of two indole alkaloids β -carboline, and show that the metabolites of healthy and infected diatom cells differ significantly. From the data it is clear that the pathogen is manipulating the algal metabolome during the completion of its life cycle. These findings are not totally unexpected and one could make a pretty robust prediction that this would be the case, but it is the first time that it has been demonstrated for an oomycete host interaction, which is nice. What is unclear from the paper is how the two compounds actively promote the infection? Does it help the pathogen itself? Is it an essential compound that is needed for its growth and propagation or does an increased production inhibit a possible defence response in the host? Does it induce particular genes (possibly involved in suppressing host defence) being up regulated or maybe the compounds are downregulating host defence genes? Does it promote the production of particular nutrients for the pathogen? Such experiments would be relatively easy to do with the model infection system that the authors have set up. It requires the treatment of the diatoms with the two isolated compounds and a couple of controls and time points and perform a

simple RNAseq experiment on the cells treated. The results would be very informative as it will give you a very clear idea what genes may be affected and what pathways are triggered that may lead to a successful infection. Since the pathogen cannot be cultured in vitro without the host (my assumption), it will be more complicated to investigate what effect the compounds might have on the pathogen's own expression profile, but it might be possible to add extra β -carboline and 4-carboxy-2,3,4,9-tetrahydro-1H- β -carboline to an infected culture and see which genes are up-down regulated in the pathogen (and host). But I appreciate that this latter experiment could be a bit messy. I think to get a better picture about the role of these compounds in the infection process, a simple RNAseq experiment, as suggested above, would greatly enhance the paper as it now reads as an unfinished (very nice) story.

#The RNAseq experiment is a good suggestion and would provide indeed useful information about the gene expression during the oomycete infection. However, the main conclusions of our article are about the mechanistic aspects of interactions triggered by newly identified chemical mediators. Our revisions thus focus on the concerns about the direct mechanistic aspects and quantification issues to improve the soundness of the main conclusions. Future studies could surely focus on biosynthesis / gene regulation aspects and include RNAseq data. However, we feel that this would be beyond the scope of our current manuscript.

Figure 1 does not have a black arrow to indicate the zoospore

#the arrow was added.

Dot missing in line 166.

Line 213. Significant”t”

The corrections were added accordingly in the text

REVIEWERS' COMMENTS:

Reviewer #1 (Remarks to the Author):

All the concerns from reviewer one were addressed accordingly and the manuscript was improved a lot. Authors also provided new data in Supporting Information to support their finding.

REVIEWERS' COMMENTS:

Reviewer #1 (Remarks to the Author):

All the concerns from reviewer one were addressed accordingly and the manuscript was improved a lot. Authors also provided new data in Supporting Information to support their finding.

We thank the reviewer for his / her helpful comments and for acknowledging the improved version.